# Implementation of a Family Skills Programme in Internally Displaced People Camps in Kachin State, Myanmar

**DOI:** 10.3390/healthcare13091090

**Published:** 2025-05-07

**Authors:** Karin Haar, Aala El-Khani, Hkawng Hawng, Tun Tun Brang, Win Mar, Zin Ko Ko Lynn, Wadih Maalouf

**Affiliations:** 1Prevention, Treatment and Rehabilitation Section, Drugs Laboratory and Scientific Support Branch, Division for Policy Analysis and Public Affairs, United Nations Office on Drugs and Crime (UNODC) Headquarters, Wagramer Strasse 5, A-1400 Vienna, Austria; karin.haar@un.org (K.H.); wadih.maalouf@un.org (W.M.); 2Division of Psychology and Mental Health, The University of Manchester, Manchester M13 9PL, UK; 3United Nations Office on Drugs and Crime (UNODC) Myanmar Office, 11(A) Maylikha Road, Mayangone Township, Yangon 11061, Myanmar; hkawng.hawng@un.org (H.H.); tun.brang@un.org (T.T.B.); win.mar@un.org (W.M.); 4United Nations Office on Drugs and Crime (UNODC) Regional Office for Southeast Asia and the Pacific, Raj Damnern Nok Ave., Bangkok 10200, Thailand; zinkoko.lynn@un.org

**Keywords:** family skills, Myanmar, internally displaced people (IDP), Strong Families, child mental health, parenting, resilience

## Abstract

Background/Objectives: Children that are forcibly displaced are more likely to experience mental health and behavioural challenges than non-displaced populations, including increased risk of anxiety, depression and post-traumatic stress disorder. Building appropriate parenting skills to strengthen the relationships between caregivers and their children in times of war is key to building resilience in children. There is a lack of research on the role of family skill interventions in internally displaced people (IDP). The aim of this study was to assess the potential change in parenting skills, child mental health, and resilience capacity in families living in IDP camps in Kachin State, Myanmar, after taking part in a brief family skill intervention, Strong Families. Methods: An open, multi-site pilot feasibility and acceptability trial was conducted with 100 families. Outcome data were collected prospectively, assessing changes in parenting skills and family adjustment in caregivers, children’s behaviour, and children’s resilience capacities. Families were assessed using three scales, Parenting and Family Adjustment Scales (PAFAS), the Strengths and Difficulties Questionnaire (SDQ), and the Child and Youth Resilience Measure (CYRM-R). Results: Despite being a light intervention, Strong Families produced improvements in the child mental health, parenting practices, and parent and family adjustment skills scales. Improvements were observed in scores particularly for caregivers and children with greater challenges at baseline. Conclusions: The results positively value the importance and feasibility of family skill interventions being integrated into the routine care of IDP families. This advocates for the prioritisation of using such tools for supporting better family functioning and mental health in humanitarian contexts.

## 1. Introduction

### 1.1. Child Wellbeing and Caregiver Support for Displaced Populations

Forced displacement impacts children and threatens their safety in many ways by exposing them to traumatic events and depriving them of multiple resources [1]. Displacement also carries indirect consequences, impacting family functioning and the care children receive from their primary caregivers [2,3]. Children that experience forced displacement are much more likely to experience mental health and behavioural challenges relative to non-displaced populations; these include increased risks of anxiety, depression, and post-traumatic stress disorder (PTSD) [4]. Building appropriate parenting skills to strengthen the relationships between caregivers and their children in times of war is key to building resilience in children [5].

A recent systematic review of 38 studies of families living in war zones identified that war-time parenting practices involved less warmth, less praise, greater hostility, and increased physical violence [6]. Previous research indicated that even caregivers that were knowledgeable regarding effective parenting skills and enjoyed a close bond with their children prior to the circumstances that led to their displacement found themselves unable to respond appropriately to their children’s emerging needs [7]. This could be due to the combination of stress caregivers experience themselves or to the novel needs their children are presenting and to the lack of pre-existing strategies or knowledge on how they might face them [8,9].

### 1.2. Family Skill Programmes

Research indicates that interventions that encourage safe and nurturing relationships between caregivers and their children can prevent several negative social outcomes including child maltreatment, poor mental health, and drug use [10]. Such family skill programmes provide caregivers with a set of new skills that allow them to cope with and adapt to the different challenges that arise while parenting children under their care. These interventions are typically delivered with opportunities for practicing these skills through competency enhancement and support, and research indicates that this practice is key to the success of such programmes [11]. Family skill interventions are recognised as primary prevention programmes that aim to strengthen the bond and attachment between caregivers and their children. They strengthen the parenting skills that build key family protective factors, including communication, trust building, problem-solving, and conflict resolution. For families experiencing displacement, strengthening parenting skills correlates to improved outcomes for both children and their caregivers [3]. Caregivers benefit too from such interventions, including reductions in PTSD and depression symptoms or improvements in standard of living [12,13].

Over the past two decades, multiple global institutions and initiatives have listed and recommended evidence-based family or parenting skill programmes as a common denominator intervention serving multiple outcomes. These include the UNODC WHO International Standards on Drug Use Prevention [14], INSPIRE initiative to end violence against children [15], the WHO-led Violence Prevention Alliance [16], and the WHO/UNICEF Helping Adolescents Thrive initiative to prevent and promote mental health in adolescence [17]. While there is much evidence of the effectiveness of such interventions in high-income and stable contexts, suggesting the potential of such programmes in other settings [18,19], the current evidence in displaced populations is evolving [20]. A recent systematic review aimed at identifying the types of caregiver or parenting interventions that have been evaluated among displaced families showed significant beneficial effects across the domains of parenting behaviours and attitudes, child psychosocial and developmental outcomes, and parent mental health [21].

The war-time threats and uncertainties that pose significant challenges to the implementation and evaluation of such programmes are driving factors for the lack of the implementation of family skill interventions for displaced families, despite the available theory and research indicating the potential benefits [22]. A further logistical impediment in the adoption of such tools is that the majority of evidence-based family skill interventions have lengthy sessions with high royalty fees. These factors lead them to being unsuitable or not prioritised for displaced populations [7]. In response, The United Nations Office on Drugs and Crime (UNODC) has been implementing a global initiative on prevention that includes piloting evidence-based family skill preventions designed and tailored for LMICs [23]. This initiative evolved into the development of the Strong Families programme as a selective, brief, family skill prevention intervention, designed to improve parenting skills, family resilience [24], as well as child well-being and family mental health per the UNODC WHO International Standards on Drug Use Prevention [14].

The Strong Families programme is aimed at families with children aged between eight and fifteen years living in low-resourced and stressful settings (including in humanitarian settings). The goal of the Strong Families programme is to support families in both recognising their strengths and skills as well as further building on their strengths by sharing their challenges and the things that work for them. Extensive information on the theoretical basis of the Strong Families programme can be found in a recent study by Haar and colleagues [25].

The Strong Families programme was first piloted in Afghanistan [26] and has since been implemented with families in over thirty countries. Research findings, so far, from RCT and single-arm implementation studies in Afghanistan [26], Serbia [27], and Iran [25] have indicated that the Strong Families programme is feasibly implemented in low-resource and challenged settings and can be delivered by training lay facilitators.

### 1.3. Aims and Objectives

The aim of this pilot study was to assess the potential change in parenting skills essential for child mental health and resilience capacity in families living in IDP camps in Kachin State through the exposure to a brief family skill intervention, Strong Families. This advocates for the prioritisation of such interventions in IDP settings in Myanmar and other similar contexts.

### 1.4. Country Context

Kachin State is the northernmost state in Myanmar, bordered by China to the north and east and India to the west. Kachin State has the highest prevalence of drug use, mainly the injecting of heroin [28]. This study site had experienced long-standing conflict, which was exacerbated by renewed fighting between Myanmar’s military and the Kachin Independence Army in 2012. Families were forced to move from their villages in Kachin State. Almost 92,500 people were forcibly displaced to live in 116 IDP camps in government-controlled areas and in 18 camps and 1 boarding house in non-government-controlled areas. Across Myanmar, over 1.4 million people are now displaced across the country, of whom more than 1.1 million have been displaced since the 2021 military takeover [29]. According to the humanitarian update provided by the United Nations Office for the Coordination of Humanitarian Affairs (OCHA) on 2 December 2022: “Armed clashes, compounded by tight security, access restrictions, and threats against aid workers, continue to hamper humanitarian operations across Myanmar” [30]. Inflation in commodity prices, including for food and fuel, deepened the socio-economic stress on communities, led many to adopt negative coping mechanisms including drug use. Nearly 31,000 civilian properties, including houses, churches, monasteries, and schools, are estimated to have been destroyed, although exact figures are difficult to verify. The level of destruction of civilian property, particularly of homes, combined with the deteriorating security situation led to delaying of the returns of people and the prolonging of people’s displacement [30]. According to a baseline assessment on the drug use among those in IDP camps and surrounding communities in three selected townships in Kachin State, the prevalence of any form of drug use was 9.35% [31]. Drug use and drug use disorders can also exacerbate the existing problems in IDP camps including parenting challenges. In addition, they can create a cycle of addiction and dependency that can be difficult to address without appropriate structured support.

Parenting practices among IDP living in Kachin State varied depending on the individual circumstances and challenges faced by each family. In general, fathers tended to focus on providing basic needs such as food, shelter, and healthcare, while mothers had a primary role of caregiving for the children in the families. However, parenting in IDP camps could also be challenging due to the lack of resources, limited access to healthcare, and the trauma of displacement. Parents also faced difficulties in balancing their responsibilities as caregivers with the need to provide for their families [32].

## 2. Materials and Methods

### 2.1. Programme Intervention

The Strong Families programme is a selective evidence-informed prevention intervention designed to improve parenting skills, child well-being, and family mental health. It is designed for families with children between 8 and 15 years. Strong Families is a group intervention, consisting of separate child and caregiver sessions, followed by joint family sessions. Caregivers are required to attend sessions over three consecutive weeks, and they are joined by their children in weeks two and three. Each session runs 1 h, meaning a total investment time for families of 7 h, as shown in Table 1, which illustrates the title of each of the caregiver, child, and family sessions that participants take and in which weeks.

### 2.2. Trial Design, Sampling, Eligibility Criteria, and Sample Size

To evaluate the programme, an open, multi-site pilot feasibility and acceptability study was conducted. While plans were initially in place to conduct a randomised control trial, news of the potential closing of the camps, which could have led to the movement of families and thus the inability to then provide the waitlist group with the intervention, shifted the operations to a pilot feasibility trial. We prospectively collected outcome data, assessing changes in parenting skills and family adjustment in caregivers, children’s behaviour, and children’s resilience capacities. Six field facilitators were selected given their level of experience in working in IDP settings. Despite their IDP engagement experience, most of the facilitators did not have a technical background on drugs and health. However, they were trained on the Strong Families programme in person in May/June 2021, and they subsequently delivered the programme to 100 families within two rounds of implementation. Prior to commencing this study in IDP camps, the six facilitators received another round of training on the Strong Families programme in February 2022. This booster training was conducted remotely by two experienced international master trainers, using an eLearning platform for remote facilitator training developed by UNODC [10].

To select IDP and families, UNODC’s field team met with IDP camp leaders and committee members from 22 IDP camps in Myitkyina, Kachin State. It was agreed to choose four IDP camps in the area that could provide space for organising the Strong Families programme with a specific number of families and were conducive for follow-up assessments of the feasibility and effectiveness of the programme. The four camps were chosen (1) because of their geographic location being within 20 miles from downtown Myitkyina, (2) because the UNODC and the field facilitators had previous experience working in these camps, and (3) it was agreed by consensus that the challenges faced, including drug use problems, within these four camps were the same as those in the remaining 18 IDP camps in Myitkyina.

The selected IDP camps were as follows:Jaw Masat IDP Camp: This camp was established with IDP from Aung Lawt, Tanai township, and was in Jaw Masat village, Myitkyina township, Kachin state, Myanmar. The camp had a total of 137 households with 657 residents. The residents were primarily farmers, and they faced difficulties in finding job opportunities in Myitkyina. They often travelled to faraway hills and mountains to cultivate crops, particularly rice. They returned to the IDP camp once a month to receive food from the World Food Programme (WFP), and some only returned every two or three months. Most parents stayed in the camp from February to May, while the children usually attended school during the government academic year.Trinity IDP camp: This camp was established in 2018 as a result of air strikes due to a clash between the Myanmar military and the Kachin Independence Army that affected 27 villages. It consisted 198 households, including 52 non-camp households, with a total population of 965. Sixty percent of parents had left the IDP camp to work in the hills and mountains for sustenance, returning only once a month to queue for the food provided by the WFP at the camp.Njan Dung IDP camp: This camp was established in 2011 due to a clash between the Myanmar military and the Kachin Independence Army. It was located in Njang Dung ward, Myitkyina township, Kachin State, Myanmar. The residents of the camps were from over 10 villages, with a significant proportion from Gara Yang village in Waimaw township. The camp hosted 73 households with over 300 residents. Approximately half of the parents worked as daily laborers within the camp vicinity, while the remaining travelled to distant locations for work, including up to the borders of China.Ziun IDP camp: This IDP camp was located in Alay Kone ward, Myitkyina township, and was established in 2011 due to a clash between the military and the Kachin Independence Army. Its residents were from 15 villages and comprised over 120 households and 600 residents. Most parents worked as daily labourers around/within the town, remaining close to their children in the camp.

All families living in the four IDP camps were invited by the respective camp management committees to the information sessions about the Strong Families programme in February/March 2022. Field facilitators conducted information meetings with family members to describe the intervention and related activities as well as to assess their willingness and availability to participate in all the programme sessions and data collection meetings. Families were voluntarily invited to attend the programme with a maximum of two of their children aged between 8 and 15 years.

To assess changes in children’s behaviour, resilience parenting skills, and family adjustment, an open, prospective, pilot feasibility trial was conducted with an embedded effectiveness evaluation using opportunistic sampling. A ‘universal’ approach was utilised, in which all families in the IDP camps were invited, not specifically targeting those with a particular known risk. Families were eligible to participate if they spoke Myanmar or Jingphaw (the language spoken by the IDP of Kachin ethnicity), were willing to take part in the programme, and were in the camps for the duration of this study and subsequent measurement meetings. We excluded families who had potentially already taken part in any other family skill programme in the past 6 months or those in which the caregiver lived separately from the child.

### 2.3. Data Collection

Caregivers provided data on demographics, emotional and behavioural difficulties as well as resilience of children, and parental and family adjustment skills through self-administered, paper-based questionnaires at three time points: (1) at baseline/pre-intervention (t1, one week before programme implementation), (2) post-intervention (t2, two weeks after the completion of the programme) and (3) at follow-up (t3, six weeks after completion of the programme). Only at t1, a standardised Family Demographic Questionnaire (FDQ) was completed, whereas the Strengths and Difficulties Questionnaire (SDQ), the Parenting and Family Adjustment Scales (PAFAS), and the Child and Youth Resilience measure (CYRM-R) were filled in at all three measurement meetings.

A similar FDQ has been used previously [25,26,27] and provided a general overview of the demographics of the caregivers and children and their living conditions.

The SDQ is a widely used and brief behavioural screening questionnaire [33] that consists of 25 items on psychological attributes, divided into 5 scales of 5 items each: (1) emotional symptoms, (2) conduct problems, (3) hyperactivity/inattention, (4) peer relationship problems, and (5) prosocial behaviour. The sum of scales (1) to (4) generates the total difficulties score (based on 20 items). Each item ranges from 0 to 2 points, resulting in a maximum of 40 points on the total difficulty score. The higher the scores on the scale (1) to (4), the higher the level of difficulties, whereas the prosocial scale is reversed, with higher scores indicating more prosocial behaviour. Children scoring ≥17 points on the total difficulties scale can be categorised into the high or very high category [34]. The SDQ has been previously used in children in Myanmar [35,36] and in Rohingya refugee children in Bangladesh [37].

The PAFAS is a 30-item self-report questionnaire for caregivers that measures two domains of parent and family functioning: the “Parenting scale”, measuring the parenting practices and quality of the parent–child relationship, and the “Family Adjustment scale”, measuring parental emotional adjustment as well as partner and family support in parenting. It was designed as a brief outcome measure to assess changes in parenting practices and parental adjustment to evaluate parenting interventions [38]. The domains are divided into seven subscales, with three to five items each, with possible scores of 0 to 3 points, resulting in different ranges of the subscales: parental consistency [0–15], coercive parenting [0–15], positive encouragement [0–9], parent–child relationship [0–15], parental adjustment [0–15], family relationships [0–12], and parental teamwork [0–9]. To categorise caregivers according to their scores at baseline, we stratified them into three layers at baseline, with highest scores representing caregivers with fewer skills and the most problems.

The CYRM-R is a 17-item self-report measure of social–ecological resilience [39]. In our study, we used the Person Most Knowledgeable (PMK) version of the child version, using a 5-point Likert scale with previously used pictorial scales [40], ranging from 1 to 5 points for each question. The 17 items are summed to gain a total score of an individual’s resilience, with overall resilience scores ranging from 17 to 85 points. Ten items are summed to calculate the personal resilience subscale (including intrapersonal and interpersonal items), ranging from 10 to 50 points, and seven items form the caregiver resilience subscale (relating to the characteristics associated with the relationships important to the primary caregiver), ranging from 7 to 35 points [41].

### 2.4. Statistical Analysis

Descriptive analyses were conducted, and associations were assessed using the chi-squared test to obtain *p*-values, using a statistical significance level of 5%. Comparisons of continuous data were performed using Student’s *t*-test. We assessed plausibility and data completeness prior to analysis and did not impute data, as it was considered valid to ignore missing data [42]. We assessed a potential group interaction effect through a two-way mixed ANOVA with within- and between-subject factors and then further tested the effects of the respective outcome variable for families in the different groups separately through repeated-measures ANOVA. We also conducted a multiple regression to determine how much of the variation in the dependent variable was explained by the independent variables. Statistical analyses were performed using SPSS (version 26; IBM, Armonk, NY, USA).

## 3. Results

### 3.1. Recruitment, Follow-Up, and Erroneous Data

Overall, 100 families participated in the programme with a follow-up of 91% at t2 and 97% at t3, as shown in Table 2.

There were no missing FDQ questionnaires, and data completeness was 100% for most questions, apart from partner’s education and partner’s work status, as expected.

Likewise, all PAFAS questions were completed by more than 95% of caregivers, apart from questions that were only applicable in case the caregiver was living with a partner. Apart from SDQ question 19, which had 8/90 missing answers at t2, all other questions were completed by more than 95% of the caregivers throughout, as was the case for all PMK-CYRM-R questions.

### 3.2. Demographics of Study Participants

Despite the small number of male caregivers in our study, we did not find any statistically significant differences between male and female caregivers, apart from education, as shown in Table 3. Overall, 99% of the caregivers stated that they had experienced war or armed conflict, and, when asked about various current difficulties in their families, 79% answered that they had insufficient food available. There were no differences between girls and boys apart from the relationship to their caregivers, such that all fathers attended with boys, whereas mothers attended with 55% of girls and 45% of boys (Table 3).

### 3.3. Parenting and Family Adjustment Skills (PAFAS)

Scores on the coercive parenting scale (F(1.786,153.597) = 79.278; *p* < 0.001), the positive encouragement scale (F(1.896,164.931) = 60.743; *p* < 0.001), and the parent–child relationship scale (F(1.747,153.753) = 44.557; *p* < 0.001) decreased significantly over time, with significant changes between before the programme and both measurements after the programme, as shown in Figure 1.

Likewise, on the family adjustment subscales, the parental adjustment (F(2,172) = 29.535; *p* < 0.001) and parental teamwork (F(2,122) = 18.411; *p* < 0.001) scores decreased significantly over time, with significant changes between before the programme and both measurements after the programme, whereas the family relationship scores decreased significantly overall (F(1.835,161.491) = 45.749; *p* < 0.001) from before to after the programme, reducing even further between t2 and t3 (Figure 1).

There was no statistically significant interaction between the different camps and time on the parental consistency, positive encouragement, parent–child relationship, parental adjustment, family relationship, or parental teamwork subscales, suggesting that the scores similarly reduced for the families regardless of camp. However, on the coercive parenting subscale, we found a significant interaction between camps and time (F(6,166) = 8.197; *p* < 0.001), with families living in the Jaw Masat, Trinity, and Ziun camps showing significant reductions in coercive parenting scores over time but not those living in the Njang Dung camp. There was also no significant difference in educational level or any of the PAFAS subscales over time, meaning that irrespective of education, scores reduced significantly over time for the caregivers. We also did not find any significant differences between male and female caregivers over time; however, the small number of fathers in our sample might limit the generalisation and interpretation of the results.

The parenting and family adjustment skill results of caregivers were grouped by score at baseline. By grouping families by their scores on each subscale at baseline, we could see highly significant interactions between the three groups and time on each subscale. Throughout, the scores of caregivers with the highest scores at baseline (caregivers with fewer skills and more problems) decreased highly significantly on all subscales, similar to those with moderate scores at baseline, whose scores decreased on five of the seven parenting and family adjustment subscales (Table 4). Apart from the parental consistency subscale and the family relationship subscales, we did not see any effect on caregivers who had low scores at baseline (high skills and few problems).

### 3.4. Strengths and Difficulties Questionnaire (SDQ)

Overall, the SDQ scores of the children decreased significantly over time as shown on the total difficulty score (F(2,134) = 42.078; *p* < 0.001), which decreased from 13.6 before the programme to 8.5 after the programme and 6.9 at t3. Likewise, the scores on all other subscales highly significantly reduced (emotional problem scale (F(2,170) = 25.044; *p* < 0.001), conduct problem scale (F(1.843,160.339) = 34.813; *p* < 0.001), hyperactivity scale (F(2,166) = 20.384; *p* < 0.001), peer problem scale (F(2,150) = 20.926; *p* < 0.001)), and the Prosocial scale (F(2,174) = 38.622; *p* < 0.001)) scores increased, as shown in Figure 2.

Overall, we did not see any interaction between the different camps and time, meaning that we saw an effect over time in each of the camps. We conducted multiple regression to determine if the total difficulty score was influenced by the education of the caregiver, the children they had to care for, and if they stated that there was insufficient food available for the family. There was however no statistical significance found in this model.

A further multiple regression (Table 5) was conducted in which we assessed how much of the variation in the “total difficulty score” was explained by the sex and age of the child. The model statistically significantly predicted the “total difficulty score” (F(2,86) = 5.811, *p* = 0.004, adj. R^2^ = 0.099). However, only the age of the child added statistically significantly to the prediction (*p* = 0.002). The model was not significant at t2 or t3, and hence the age of the child did not add to the prediction after the programme.

In children with high or very high scores (≥17 points) at baseline and who completed all three data collection points (10 boys and 7 girls), we found highly significant reductions in scores in both genders, from 19.6 overall at baseline to 10.3 at time 2 and 7.4 at time 3, as shown in Table 6.

Overall, there was no interaction between the sex of the child and time on the total difficulty scores, and the scores of both girls and boys decreased significantly over time, from 13.1 to 7.5 and 7.3 for girls (F(2,70) = 22.780; *p* < 0.001) and 14.3 to 9.6 and 6.6 for boys (F(2,62) = 21.496; *p* < 0.001).

### 3.5. Child and Youth Resilience Measure (CYRM-R)

The total CYRM-R scores increased (improved) significantly over time from 59.2 before the programme to 72.1 after the programme and 72.8 at t3 (F(1.619,129.538) = 90.254; *p* < 0.001). Likewise, the personal and caregiver resilience subscale scores increased highly significantly over time, as shown in Figure 3.

On the overall scale, we found an interaction between the different camps and time (F(5.445,139.754) = 5.128; *p* < 0.001), with significant increases in the scores in the Jaw Masat (F(1.684,55.562) = 84.908; *p* < 0.001), Trinity (F(2,62) = 21.312; *p* < 0.001), and Ziun (F(2,14) = 14.952; *p* < 0.001) camps but not the Njang Dung camp. This was due to highly significant changes on both subscales in the three camps apart from the Njang Dung camp.

As for the resilience results of the children split according to their scores at baseline, although significant in all three groups, the scores of children with low scores at baseline (17–55 points on the total CYRM-R scale) increased highly significantly after the programme, increasing from 47 points on average before to 71 points after the programme and 72 points at t3 (Table 7).

Overall, there was no effect of sex on changes in the scores over time on the total CYRM-R, and the scores of boys and girls similarly increased significantly (F(1.313,47.251) = 29.134; *p* < 0.001 and F(2,86) = 66.738; *p* < 0.001), from 60.6 to 72.6 and 73.3 for boys and 58 to 71.6 and 72.3 points for girls.

## 4. Discussion

### 4.1. Overall Effect of the Strong Families Programme

The results of this study suggest that the skills of the families living in IDP camps in Kachin State significantly improved across all scales used in this analysis following their exposure to this short family skill intervention (Strong Families). Despite being a light intervention, Strong Families produced improvement in child mental health, parenting practices, and parent and family adjustment skills.

Although we did not include a comparison group, it was noticeable that the improvements in scores across scales were more pronounced in the sub-groups of caregivers and children with greater challenges at baseline. This agrees with the results noted in previous implementations of Strong Families in LMICs [25,26,27] and supports the strategy of recruiting families on a universal level rather than on a selective level. Such a modality gives the option of including everyone, which is not only easier to implement but also less stigmatising to any families with potentially unrecognised problems. Nevertheless, Strong Families is a preventive group intervention and is not intended or designed for families with severe mental health problems. Facilitators are advised during training to observe families within their first encounter and, if needed, refer them to more specialised care in their vicinity.

Almost all families included in this study reported that they had experienced war or armed conflict; however, we did not ask when and hence could not verify if they were able to overcome any past traumatising events (perhaps even with external help), if these events occurred recently, or if they had trauma symptoms. Furthermore, we did not specifically ask how stressful the living conditions in the camps were; although, from the responses to the potential current difficulties within their families, where 79% answered that they had insufficient food available and 35% were lacking job opportunities, we can postulate challenging conditions. In addition, we almost exclusively had female caregivers attending the programme. This is a common occurrence in family skill intervention implementation, in which female caregivers frequently attend as the primary caregivers [25,27]. This was particularly true in the case of this study’s participants, as the majority of the fathers worked away from the living spaces and were not often present for extended periods of time. Thus, it is plausible that the main reason for not attending such programmes would be the unavailability of the male caregiver rather than unwillingness or cultural norms. As men were often outside the camps, women frequently were left to raise children alone, and it is not surprising that almost 70% answered that they were not working (outside the house), and 60% had primary school education or below. That said, much could be implemented to improve participation amongst fathers in future implementations, such as working with male caregivers to identify suitable times and locations to run such interventions.

On average, while remaining within the target age range of the intervention, the children taking part in this study were on average older (12 years old) than in previous global trials of Strong Families. This average was 9.6 years in Afghanistan [26], 10.5 years in Serbia [27], and 9.7 years in Iran [25]. Despite the age average difference relative to other sites, much like the previous trials in other countries, the scores on all scales improved highly significantly in children, particularly in those with higher or moderate problems at the start point. In this study, the total difficulty score started from an average of 13.6 points, which was much lower when compared to the other three countries (17.8 in Afghanistan [26], 14.2 in Afghan children living in refugee reception centres in Serbia [27], and 16.6 in children of Afghan origin living in Iran [25]. Other differences from previous implementation sites were the contexts the families were living in, as almost all caregivers had experienced war or armed conflict, and many had insufficient food. Thus, we checked if the total difficulty score was influenced by the education of the caregiver, the number of children they had to care for, or if there was insufficient food available for the family, but there was no effect of these factors; however, the age of the child predicted the “total difficulty score” in another model. We, therefore, recommend further research exploring the age and maturity of children and the benefits they obtain through family skills programmes.

Even though children started with a total difficulty score of 13.6 at baseline, which, per SDQ standards, indicated that they did not have any problems on average [33], their score still decreased (improved) significantly after exposure to the Strong Families programme. This was noted in the total difficulty score, as well as for all individual SDQ subscales. Similar to the other scales, in children with high or very high scores (≥17 points) at baseline, we found highly significant reductions in scores in both sexes. While an RCT is still recommended to confirm these results, these are nevertheless reassuring and encouraging results. Furthermore, we did not find any influence of sex on the reduction in the SDQ scores amongst those exposed to Strong Family. This aligns with what we postulated previously: Strong Families has an effect on both sexes in children [25]. These results also support the advocacy of such parenting packages to support Sustainable Development Goal 5 (Gender Equality) despite the challenges in such settings.

Overall, the observation that the scores of the caregivers decreased significantly on six out of the seven PAFAS subscales, and that this decrease was sustained six weeks after the programme was received, is also very encouraging. Furthermore, we did not find an effect of education of the caregiver or their sex on the PAFAS score reduction. This shows the lack of sex sensitivity of the tool not only to children but also to caregivers. Nevertheless, despite encouraging results, such sex sensitivity findings need to be interpreted with caution as they need to be further corroborated by a randomised clinical trial (and through a research protocol ensuring a higher number of fathers).

Similarly, the same results were found for the child resilience measures, such as improvement over time, children with higher (worse) scores at baseline improving most (though all improved too), and girls and boys improving likewise. This observation was noted across all camps except Njang Dung camp, where children did not obtain a similar improvement as those in the other camps. This was also noted by the scores on the coercive parenting subscale not improving as significantly in the Njang Dung camp. Despite the small sample size of the families reached in Njang Dung camp (total of 10), we endeavoured to explore the cause of this peculiar trend. Our investigation revealed that Njang Dung camp is an older and established camp (established in 2011) as compared to the other camps, which were established in 2018. Furthermore, according to the qualitative feedback from our facilitators, the children in this camp are notably more educated and proficient in communication, having obtained other psychosocial support in the past. This implies that their initial level of competence was higher.

Overall, we saw very high follow-up rates and very little missing data in our sample. This may have been partly due to the activities of the research staff who were responsible for the data collection. They had extensive experience conducting research with local families and were also trained remotely on the evaluation tools used in this study. In addition, the research assistants were in close contact with camp leaders and had informed all families on the exact dates of sessions ahead of the initiation of the trial. For future impact evaluations and longer-term follow-up studies, this experience is valuable and calls for the future capacity building of the evaluators in other countries.

### 4.2. Limitations

Despite the positive results of this study, the main limitation of our study is that no control group was included to verify the effects of Strong Families on the aforementioned indicators. While the results indicate statistically significant improvements in parenting skills, child mental health, and resilience, this study did not account for external factors that may have contributed to these changes, such as seasonal variations, additional humanitarian aid, or natural child development. Without a control group, the improvements may not be directly attributable to the intervention; thus, in the future, we will include a matched waitlist group to allow for thorough comparison.

It is important to note that some IDP camps showed larger improvements than others. For example, the effects were weaker for Njang Dung camp. These differences might have been due to a number of reasons such as potential contextual factors, such as prior exposure to psychosocial support, amount of income (if any), education level, and camp conditions. These factors were not accounted for in the research tools used in this study. In our upcoming trials, we aim to include additional contextual as well as demographic indicators, such as history of trauma/armed events, length of and reason for stay in the camps, as well as what support has already been provided to the families. The addition of a qualitative research arm in further studies to obtain an in-depth understanding of participants’ experiences, perceptions and social contexts is also planned in the upcoming RCT of Strong Families in Myanmar refugee camps.

We also aim to include a longer-term follow-up to be better able to comment on the sustainability of the effects of our programme. Future research is essential to assess the long-term impact of the programme and to compare children with or without receiving intervention through a thorough outcome and impact evaluation. Longer follow-up times are often challenging in humanitarian contexts, particularly within the context of refugee instability, where refugees may move soon after starting participation.

Our initial results can be used to motivate policy makers to allow for such an evaluation and integrate such programmes into their country’s implementation strategies for reducing negative health and social outcomes. Finally, further verification would be beneficial to explore how representative the four selected camps were compared to all IDP camps in Kachin State or even the whole of Myanmar through expansion to other camps and geographical areas.

## 5. Conclusions

The implementation of a brief family skill group intervention was found to be feasible in IDP camps in Kachin State, and the results are very encouraging in terms of potential benefit to children and their caregivers. The fact that those with greater challenges or fewer skills prior to taking part in the intervention seemed to have benefited more from the intervention is particularly reassuring. However, to verify the effects of the Strong Families programme, a future trial that includes a comparison group is recommended to further corroborate the findings. Further data collection on the long-term effects, generalisability, and the role of male caregivers could be obtained through expansion to other camps in the area, the use of a longer follow-up time, and the inclusion of male caregivers in the programme. To the best of our knowledge, the previously trained facilitators have in the meantime continued to deliver the Strong Families programme to families in IDP camps in the area; however, due to funding restraints, they were not able to include a research component. It is, however, very reassuring that the programme could be integrated into their routine care for families to ensure sustainability in the future.

## Figures and Tables

**Figure 1 healthcare-13-01090-f001:**
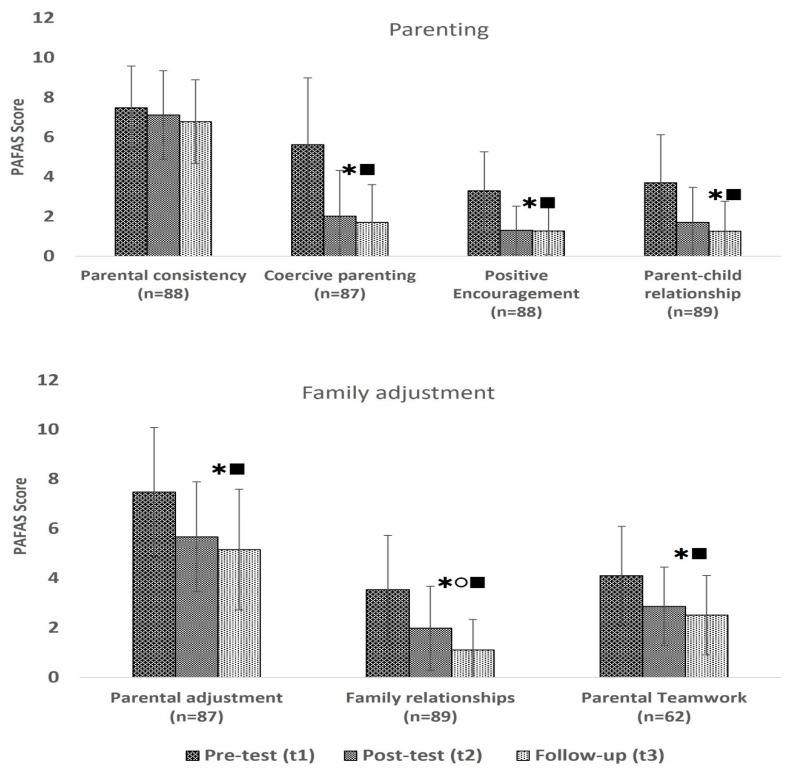
Mean PAFAS scores (+/−standard deviation) over time; 
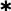
 significant difference between t1 and t2, 
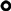
 significant difference between t2 and t3, 
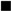
 significant difference between t1 and t3.

**Figure 2 healthcare-13-01090-f002:**
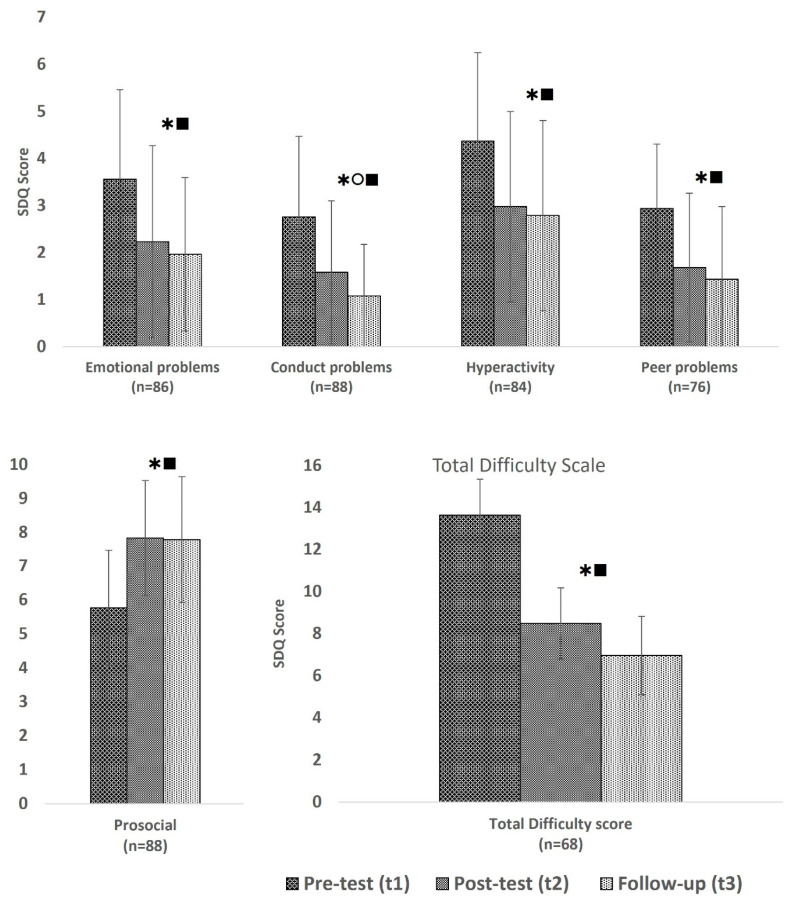
Mean SDQ scores (+/−standard deviation) over time; 
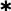
 significant difference between t1 and t2, 
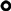
 significant difference between t2 and t3, 
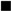
 significant difference between t1 and t3.

**Figure 3 healthcare-13-01090-f003:**
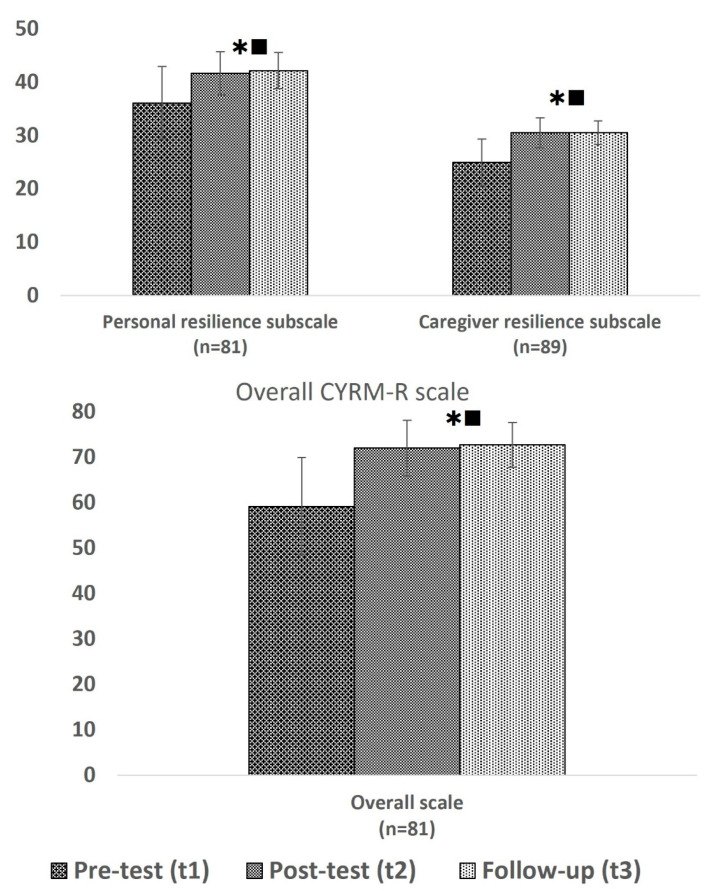
Overall CYRM-R scale as well as personal and caregiver resilience subscales before and after the programme: 
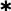
 significant difference between t1 and t2, 
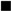
 significant difference between t1 and t3.

**Table 1 healthcare-13-01090-t001:** Structure of the Strong Families programme [8].

Week 1	Week 2		Week 3	
Caregiver pre-session Understanding Strengths and Stresses	Caregiver session 1 Using Love and Limits	In parallel	Caregiver session 2 Teaching children what is right	In parallel
	Child session 1 Learning about stress		Child session 2 Following rules and appreciating caregivers	
	Family session 1 Learning about each other		Family session 2 Supporting values and dreams	

**Table 2 healthcare-13-01090-t002:** Follow-up of families by camp.

IDP Camp	Pre-Intervention t1	Post-Intervention t2	Follow-Up t3
Jaw Masat n = 40	40 100%	39 98%	39 98%
Njang Dung n = 10	10 100%	8 80%	10 100%
Trinity n = 40	40 100%	34 85%	38 95%
Ziun n = 10	10 100%	10 100%	10 100%
Total n = 100	100 100%	91 91%	97 97%

**Table 3 healthcare-13-01090-t003:** Demographic characteristics of caregivers and children in Kachin State, Myanmar.

Caregiver Demographics		Total (n = 100)	Female Caregiver (n = 96)	Male Caregiver (n = 4)	*p*-Value	Chi^2^, *t*-Test
		Mean (SD); n (%)	Mean (SD); n (%)	Mean (SD); n (%)		
Age (in years)		41.0 (10.28)	40.8 (10.19)	45.8 (12.84)	0.352	t_98_ = −0.935
Marital status	Married	80 (80%)	77 (80%)	3 (75%)	0.858	Χ^2^ = 0.766
	Divorced/separated	5 (5%)	5 (5%)	-		
	Single	2 (2%)	2 (2%)	-		
	Widow	13 (13%)	12 (13%)	1 (25%)		
Education	Primary school or less	60 (60%)	58 (60%)	2 (50%)	<0.001	Χ^2^ = 24.315
	Some high school	37 (37%)	36 (38%)	1 (25%)		
	Completed high school	1 (1%)	-	1 (25%)		
	University degree	2 (2%)	2 (2%)	-		
Partner’s education	Primary school or less	59 (69%)	57 (79%)	2 (67%)	0.972	Χ^2^ = 0.056
	Some high school	25 (29%)	24 (29%)	1 (33%)		
	Completed high school	1 (1%)	1 (1%)	-		
Work status	Full time	4 (4%)	4 (4%)	-	0.759	Χ^2^ = 1.872
	Part time	8 (8%)	8 (8%)	-		
	Not working but looking for a job	5 (5%)	5 (5%)	-		
	Home based paid work	14 (14%)	14 (15%)	-		
	Not working	69 (69%)	65 (68%)	4 (100%)		
If working, what kind of work	Voluntary work (unpaid)	3 (20%)	3 (20%)	-	n/a	n/a
	Voluntary work (paid salary monthly but counting daily basis)	1 (7%)	1 (7%)	-		
	Cash for Work (daily basis labour)	7 (47%)	7 (47%)	-		
	Others	4 (27%)	4 (27%)	-		
Partner’s work status	Full time	34 (42%)	33 (42%)	1 (33%)	0.723	Χ^2^ = 1.327
	Part time	31 (38%)	29 (37%)	2 (67%)		
	Not working but looking for a job	2 (3%)	2 (3%)	-		
	Not working	14 (17%)	14 (18%)	-		
If the partner is working, what kind of work	Voluntary work (unpaid)	2 (3%)	2 (3%)	-	0.495	Χ^2^ = 2.390
	Voluntary work (paid salary monthly but counting daily basis)	6 (9%)	5 (8%)	1 (33%)		
	Cash for work (daily basis labour)	37 (57%)	36 (58%)	1 (33%)		
	Others	20 (31%)	19 (31%)	1 (33%)		
Current difficulties in the family (Multiple answer question)	Insufficient food	77 (79%)	74 (79%)	3 (75%)		
	Lack of job opportunities	34 (35%)	32 (34%)	2 (50%)		
	Insufficient drinking water	9 (9%)	9 (10%)	-		
	Quarrelling with partner	3 (3%)	3 (3%)	-		
	Other (no income, education, lack of school fees, struggling alone, caring for chronic patients, poor health, lack of healthcare, narrow space, disabled child, unable to afford children’s needs, etc.)	48 (49%)	47 (50%)	1 (25%)		
Experienced war or armed conflict in the past	Yes	99 (99%)	95 (99%)	4 (100%)	0.837	Χ^2^ = 0.042
	No	1 (1%)	1 (1%)	-		
Number of children		3.0 (1.33)	3.0 (1.35)	2.8 (0.50)	0.669	t_98_ = −0.428
Child demographics		Total (n = 100)	Girls (n = 54)	Boys (n = 46)	*p*-value	Chi^2^, *t*-test
Age of child taking part in the programme (in years)		12.1 (2.37)	11.8 (2.26)	12.4 (2.48)	0.245	t_98_ = 1.170
Relationship of the caregiver to the child	Mother	84 (84%)	46 (55%)	38 (45%)	0.025	Χ^2^ = 12.871
	Father	4 (4%)	-	4 (100%)		
	Grandmother	6 (6%)	5 (83%)	1 (17%)		
	Sister	3 (3%)	-	3 (100%)		
	Stepmother	1 (1%)	1 (100%)	-		
	Other	2 (2%)	2 (100%)	-		

**Table 4 healthcare-13-01090-t004:** PAFAS of caregivers categorised by their pre-test scores.

	Pre-Test Family Scores	Pre-TestMean (SD)	Post-TestMean (SD)	Follow-UpMean(SD)	Two-Way Mixed ANOVAF(df_time_, df_error_); *p*-Value	Repeated-Measures ANOVAF(df_time_, df_error_); *p*-Value	Post Hoc Tests
PARENTING						
Parental Consistency	≥9 (n = 31)	9.77 (0.92)	6.65 (2.21)	7.45 (1.91)	F(3.821,162.389) = 16.503; *p* < 0.001	F(2,60) = 24.999; *p* < 0.001	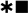
7–8 (n = 29)	7.34 (0.48)	8.10 (2.14)	6.72 (2.21)	F(2,56) = 4.379; *p* = 0.017	
≤6 (n = 28)	5.07 (1.05)	6.64 (2.08)	6.11 (2.06)	F(2,54) = 6.149; *p* = 0.004	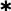
Coercive Parenting	≥7 (n = 34)	9.18 (1.59)	3.00 (2.89)	2.15 (1.83)	F(4,168) = 33.055; *p* < 0.001	F(2,66) = 112.546; *p* < 0.001	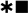
4–6 (n = 28)	4.89 (0.83)	1.39 (1.66)	1.93 (2.02)	F(2,54) = 44.419; *p* < 0.001	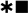
≤3 (n = 25)	1.60 (1.19)	1.44 (1.47)	0.92 (1.68)		
Positive Encouragement	≥4 (n = 38)	5.38 (1.16)	1.38 (1.26)	1.41 (1.28)	F(3.795,161.272) = 36.106; *p* < 0.001	F(2,66) = 135.037; *p* < 0.001	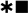
3(n = 25)	3.00 (0)	1.45 (1.34)	1.73 (1.39)	F(2,42) = 12.903; *p* < 0.001	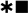
≤2 (n = 35)	1.31 (0.74)	1.13 (1.13)	0.84 (0.81)		
Parent–child Relationship	≥5 (n = 31)	6.32 (1.51)	2.13 (1.93)	1.19 (1.60)	F(4,172) = 31.134; *p* < 0.001	F(2,60) = 77.938; *p* < 0.001	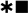
3–4 (n = 30)	3.50 (0.51)	1.47 (1.43)	1.40 (1.57)	F(2,58) = 31.312; *p* < 0.001	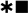
≤2 (n = 28)	1.04 (0.92)	1.54 (1.88)	1.18 (1.42)		
FAMILY ADJUSTMENT						
Parental Adjustment	≥9 (n = 29)	10.48 (1.27)	6.28 (2.05)	5.48 (2.59)	F(4,168) = 17.563; *p* < 0.001	F(2,56) = 50.008; *p* < 0.001	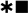
7–8 (n = 24)	7.42 (0.50)	5.71 (1.90)	5.33 (2.22)	F(2,46) = 12.827; *p* < 0.001	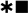
≤6 (n = 34)	4.97 (1.36)	5.15 (2.45)	4.76 (2.45)		
Family relationships	≥5 (n = 27)	6.15 (1.59)	2.07 (2.04)	1.30 (1.44)	F(3.787,162.848) = 23.158; *p* < 0.001	F(2,52) *= 50.185*; *p* < 0.001	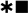
3–4 (n = 31)	3.45 (0.51)	2.32 (1.68)	1.29 (1.31)	F(1.534,46.016) = 27.555; *p* < 0.001	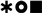
≤2 (n = 31)	1.35 (2.19)	1.58 (1.31)	0.77 (1.09)	F(2,60) = 4.786; *p* = 0.012	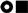
Parental teamwork	≥5 (n = 28)	5.86 (1.04)	3.11 (1.45)	2.57 (1.53)	F(4,118) = 16.540; *p* < 0.001	F(2,54) = 62.948; *p* < 0.001	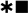
3–4 (n = 20)	3.65 (0.49)	3.00 (1.86)	2.70 (1.53)		
≤2 (n = 14)	1.29 (0.73)	2.21 (1.31)	2.14 (1.92)		

*Statistically significant* (*p* < 0.05); results for repeated-measures ANOVA and post hoc tests only shown if significant; SD: standard deviation; 
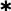
 significant difference between t1 and t2, 
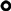
 significant difference between t2 and t3, 
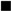
 significant difference between t1 and t3.

**Table 5 healthcare-13-01090-t005:** Multiple regression coefficients and standard errors for total difficulty score before the programme.

Total Difficulty Score	B	95% CI for B		SE B	β	R^2^	ΔR^2^
		LL	UL				
Model						0.119	0.099 *
Constant	23.29 *	17.456	29.125	2.935			
Age of the child	−0.613 *	−1.002	−0.223	0.196	−0.319		
Sex of the child	−1.654	−3.487	0.180	0.922	−0.183		

Note. Model = “Enter” method; B = unstandardised regression coefficient; CI = confidence interval; LL = lower limit; UL = upper limit; SE B = standard error of the coefficient; β = standardised coefficient; R^2^ = coefficient of determination; ΔR^2^ = adjusted R^2^. * *p* < 0.05.

**Table 6 healthcare-13-01090-t006:** Total difficulty scores of boys and girls with 17 or more pre-test points.

SDQ	Pre-TestMean (SD)	Post-TestMean (SD)	Follow-UpMean(SD)	Two-Way Mixed ANOVAF(df_time_, df_error_); *p*-Value	Repeated-Measures ANOVAF(df_time_, df_error_); *p*-Value	Post Hoc Tests
Total Difficulty Scores in Children with 17 or Pre-Test More Points (High and Very High)
Boys (n = 10)	19.5 (2.27)	10.5 (7.29)	6.5 (3.78)	F(2,30) = 0.230; *p* = 0.796	F(2,18) = 18.359; *p* < 0.001	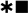
Girls (n = 7)	19.9 (3.02)	10.0 (7.83)	8.6 (8.06)	F(2,12) = 6.500; *p* = 0.012	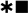

SD: standard deviation; 
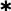
 significant difference between t1 and t2, 
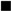
 significant difference between t1 and t3.

**Table 7 healthcare-13-01090-t007:** CYRM-R scores in children categorised by their pre-test scores.

	Pre-Test Family Scores	Pre-TestMean (SD)	Post-TestMean (SD)	Follow-UpMean(SD)	Two-Way Mixed ANOVAF(df_time_, df_error_); *p*-Value	Repeated-Measures ANOVAF(df_time_, df_error_); *p*-Value	Post Hoc Tests
Total CYRM-R scale	17–55 (n = 28)	46.89(5.59)	71.0(5.48)	71.96(5.80)	F(4,156) = 44.06; *p* < 0.001	F(2,54) = 174.885; *p* < 0.001	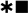
56–65 (n = 26)	60.69(2.62)	71.69(6.89)	73.27(3.94)	F(1.495,37.384) = 49.986; *p* < 0.001	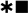
66–85 (n = 27)	70.59(3.52)	73.59(5.92)	73.15(4.99)	F(2,52) = 3.816; *p* = 0.028	
Personal resilience subscale	10–31 (n = 26)	26.08(3.79)	40.5(4.07)	42.0(3.51)	F(4,156) = 34.935; *p* < 0.001	F(2,50) = 124.339; *p* < 0.001	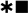
32–37(n = 26)	34.5(1.68)	41.58(3.91)	41.96(3.21)	F(1.659,41.486) = 45.156; *p* < 0.001	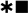
38–50 (n = 29)	40.93(2.91)	42.79(3.99)	42.48(3.46)	n.s.	
Caregiver resilience subscale	7–22 (n = 30)	19.97(2.16)	30.57(2.54)	30.17(2.78)	F(4,172) = 46.731; *p* < 0.001	F(2,58) = 214.782; *p* < 0.001	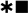
23–27 (n = 27)	25.04(1.37)	30.15(2.90)	30.56(1.76)	F(1.733,45,053) = 55.485; *p* < 0.001	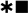
28–35 (n = 32)	29.53(1.54)	30.72(3.02)	30.91(1.99)	F(1.721,53.353) = 4.172; *p* = 0.026	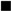

n.s.: not significant; SD: standard deviation; 
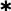
 significant difference between t1 and t2, 
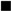
 significant difference between t1 and t3.

## Data Availability

The datasets generated and analysed during the current study are available from the Mendeley repository.

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
