# Peer review of "Implementation of a Family Skills Programme in Internally Displaced People Camps in Kachin State, Myanmar"

_healthcare, 2025, doi:10.3390/healthcare13091090_

Round 1
Reviewer 1 Report
Comments and Suggestions for Authors
The manuscript presents an important study on the implementation of the Strong Families intervention in IDP camps in Myanmar. The research topic is relevant, and the use of standardized assessment tools strengthens the study’s credibility. However, the manuscript requires substantial improvements before it can be considered for publication.
One of the most significant methodological concerns is the absence of a control group. While the results indicate statistically significant improvements in parenting skills, child mental health, and resilience, the study does not account for external factors that may have contributed to these changes, such as seasonal variations, additional humanitarian aid, or natural child development. The manuscript suggests causality in multiple instances, for example, when stating that “the results of this study suggest that families living in IDP camps significantly benefited from a short family skills intervention.” However, without a control group, the improvements may not be directly attributable to the intervention. The recommendations for integrating family skills programs into humanitarian settings are valuable, but they must be supported by stronger methodological evidence. The authors should explicitly acknowledge this limitation in both the methods and discussion sections. Ideally, future studies should include a randomized controlled trial or at least a matched comparison group.
The authors chose a pilot feasibility study, which is valid for exploratory research. However, the manuscript does not explain why this specific design was chosen over other methods. What is e.g. the advantage over a qualitative research method? Additionally, the study does not address why some IDP camps showed stronger improvements than others. For example, Njang Dung camp exhibited weaker effects, yet no discussion is provided on why this may have occurred. The authors should consider potential contextual factors, such as prior exposure to psychosocial support, education levels, and camp conditions, that might explain these variations.
While the manuscript presents p-values for statistical significance, it does not report effect sizes, such as Cohen’s d or partial eta squared. Without effect sizes, it is difficult to judge whether the observed improvements are meaningful in real-world settings. Including effect sizes in the results section would provide a more comprehensive statistical interpretation.
The manuscript requires major revisions before it can be considered for publication. The authors should acknowledge the limitations of not having a control group and avoid making causal claims, provide stronger justification for using a pilot feasibility design, discuss why results varied across IDP camps, correct typographical and grammatical errors, fix missing figure references, and include effect sizes in statistical reporting. If these concerns are addressed, the study will be much stronger and more convincing to readers.
Comments on the Quality of English LanguageSeveral typographical and grammatical errors throughout the manuscript need correction (e.g. “caregievrs” instead of “caregivers,” “appropraitly” instead of “appropriately,” “interevntions” instead of “interventions.”).
Author Response
Response to Reviewer 1 Comments
|
||
1. Summary |
|
|
Reviewer comment: The manuscript presents an important study on the implementation of the Strong Families intervention in IDP camps in Myanmar. The research topic is relevant, and the use of standardized assessment tools strengthens the study’s credibility. However, the manuscript requires substantial improvements before it can be considered for publication.
Author response: Thank you very much for taking the time to review this manuscript. Please find the detailed responses below and the corresponding revisions/corrections highlighted/in track changes in the re-submitted files.
Reviewer comment:
One of the most significant methodological concerns is the absence of a control group. While the results indicate statistically significant improvements in parenting skills, child mental health, and resilience, the study does not account for external factors that may have contributed to these changes, such as seasonal variations, additional humanitarian aid, or natural child development.
The manuscript suggests causality in multiple instances, for example, when stating that “the results of this study suggest that families living in IDP camps significantly benefited from a short family skills intervention.” However, without a control group, the improvements may not be directly attributable to the intervention. The recommendations for integrating family skills programs into humanitarian settings are valuable, but they must be supported by stronger methodological evidence. The authors should explicitly acknowledge this limitation in both the methods and discussion sections. Ideally, future studies should include a randomized controlled trial or at least a matched comparison group.
Author response:
Thank you for pointing this out. We had already mentioned in the discussion section the lack of a control group as a major limitation of the study and also reflected upon this in the conclusion. Following your comment, we have now expanded this section and also added this limitation in the method section as suggested.
Reviewer comment:
The authors chose a pilot feasibility study, which is valid for exploratory research. However, the manuscript does not explain why this specific design was chosen over other methods. What is e.g. the advantage over a qualitative research method?
Author response: We have now expanded on this in the limitation section, focusing on the benefit and further angles a qualitative arm will add. We now mention this is being added to the upcoming RCT of the intervention in Myanmar.
Reviewer comment:
Additionally, the study does not address why some IDP camps showed stronger improvements than others. For example, Njang Dung camp exhibited weaker effects, yet no discussion is provided on why this may have occurred. The authors should consider potential contextual factors, such as prior exposure to psychosocial support, education levels, and camp conditions, that might explain these variations.
Author response: Thank you for noting this. We had touched on this briefly in the limitation section already, referring to the importance of more extensive data collection to amount for these factors in the next research study. We have now expanded this in the limitation section following your very useful comment!
Reviewer comment:
While the manuscript presents p-values for statistical significance, it does not report effect sizes, such as Cohen’s d or partial eta squared. Without effect sizes, it is difficult to judge whether the observed improvements are meaningful in real-world settings. Including effect sizes in the results section would provide a more comprehensive statistical interpretation.
Author response:
We have now included effect sizes in the results section.
Reviewer comment:
The manuscript requires major revisions before it can be considered for publication. The authors should acknowledge the limitations of not having a control group and avoid making causal claims, provide stronger justification for using a pilot feasibility design, discuss why results varied across IDP camps, correct typographical and grammatical errors, fix missing figure references, and include effect sizes in statistical reporting. If these concerns are addressed, the study will be much stronger and more convincing to readers.
Author response:
We have now covered all the comments made. Thanks for your attention and time reviewing our manuscript.
Reviewer 2 Report
Comments and Suggestions for Authors
I have read the article "Implementation of Strong Families in IDP camps in Kachin State, Myanmar" with interest. This is an important contribution to the field of humanitarian intervention and child mental health, particularly in the context of internally displaced populations.
This is significant because it addresses a gap in research on how parenting interventions can improve resilience and mental health in humanitarian settings, particularly in Myanmar. The findings suggest that the Strong Families program had a positive impact on parenting skills, child mental health, and resilience among IDP families, highlighting its potential for integration into routine care.
However, while the article presents a compelling case for the effectiveness of the intervention, it lacks a control group, which limits the ability to draw causal inferences. The absence of a comparison group makes it difficult to determine whether improvements observed were solely due to the intervention or other external factors.
Also, while the study discusses gender participation, there is limited analysis of how the involvement of male caregivers—or lack thereof—might have influenced the outcomes.
The methodology, which includes an open, multisite pilot feasibility and acceptability trial, is fine to me for a preliminary assessment. The use of validated scales such as PAFAS, SDQ, and CYRM-R adds credibility to the study. However, the lack of long-term follow-up beyond six weeks weakens claims of sustained impact.
To me, the study also does not account for potential confounding variables, such as prior exposure to psychosocial support programs, which may have influenced the results. The analysis is generally sound, demonstrating statistically significant improvements, but it would have been strengthened by a mixed-methods approach that included qualitative analysis on participant experiences.
The conclusion is strong as it aligns with the findings but overstates the effectiveness of the intervention without acknowledging the need for a more rigorous study design.
Despite some methodological flaws (to me), the novelty lies in its application of a structured family skills program within an under-researched humanitarian context.
Comments on the Quality of English Language
As I rated earlier that there are rooms to improve
Author Response
Response to Reviewer 2 Comments
|
||
1. Summary |
|
|
Thank you very much for taking the time to review this manuscript. Please find the detailed responses below and the corresponding revisions/corrections highlighted/in track changes in the re-submitted files.
Reviewer comment:
I have read the article "Implementation of Strong Families in IDP camps in Kachin State, Myanmar" with interest. This is an important contribution to the field of humanitarian intervention and child mental health, particularly in the context of internally displaced populations.
This is significant because it addresses a gap in research on how parenting interventions can improve resilience and mental health in humanitarian settings, particularly in Myanmar. The findings suggest that the Strong Families program had a positive impact on parenting skills, child mental health, and resilience among IDP families, highlighting its potential for integration into routine care.
Author response:
Thank you very much for taking the time to review this manuscript. Please find the detailed responses below and the corresponding revisions/corrections highlighted/in track changes in the re-submitted files.
Reviewer comment:
However, while the article presents a compelling case for the effectiveness of the intervention, it lacks a control group, which limits the ability to draw causal inferences. The absence of a comparison group makes it difficult to determine whether improvements observed were solely due to the intervention or other external factors.
Author response:
Thank you for pointing this out. We had already mentioned in the discussion section the lack of a control group as a major limitation of the study and also reflected upon this in the conclusion. Following your comment, we have now expanded this section and also added this limitation in the method section as suggested.
Reviewer comment:
Also, while the study discusses gender participation, there is limited analysis of how the involvement of male caregivers—or lack thereof—might have influenced the outcomes.
Author response: Thank you for noting this. We have now expanded the discussion on the caregiver measure to add: This reflected that the intervention had an effect on both genders, however, the number of fathers was small, so gender-based results in caregivers need to be interpreted with caution.
We have also expanded on this topic further in the limitation. Our upcoming RCT of the intervention in Myanmar is taking your point into consideration too-thanks!
Reviewer comment:
The methodology, which includes an open, multisite pilot feasibility and acceptability trial, is fine to me for a preliminary assessment. The use of validated scales such as PAFAS, SDQ, and CYRM-R adds credibility to the study. However, the lack of long-term follow-up beyond six weeks weakens claims of sustained impact.
Author response:
We have now expanded further in the limitation section to elaborate the existing point on the need for longer follow up times and the challenges this poses in humanitarian research with movement of families.
Reviewer comment:
To me, the study also does not account for potential confounding variables, such as prior exposure to psychosocial support programs, which may have influenced the results. The analysis is generally sound, demonstrating statistically significant improvements, but it would have been strengthened by a mixed-methods approach that included qualitative analysis on participant experiences.
Author response:
Thank you for noting this. We had touched on this briefly in the limitation section already, referring to the importance of more extensive data collection to amount for these factors in the next research study. We have now expanded this in the limitation section following your very useful comment!.
Regarding mixed methods-We have now expanded on this in the limitation section, focusing on the benefit and further angles a qualitative arm will add. We now mention this is being added to the upcoming RCT of the intervention in Myanmar.
Reviewer comment:
The conclusion is strong as it aligns with the findings but overstates the effectiveness of the intervention without acknowledging the need for a more rigorous study design. Despite some methodological flaws (to me), the novelty lies in its application of a structured family skills program within an under-researched humanitarian context.
Author response:
We have now covered all the comments made. Thanks for your attention and time reviewing our manuscript.
Reviewer 3 Report
Comments and Suggestions for Authors
1-)you can improve the title of the study.
2-)write non-abbreviated version of IDP in the abstract.
3-)you can be careful about punctuation.
4-)if possible you can give more information about the results in the abstract section.
5-)you can discuss the findings (line 165) in the context of your study.
6-)213 reference is missing
7-)table 1 seems unclear.
8-)you can give more information about the participants of the study.
9-)line 340, 341 reference seems missing.
10-)you can highlight the novelty of your study in the discussion section based on the results of your study.
6-)you can give more information about how much they earn monthly and mention as a confounding factor in the study.
Comments on the Quality of English Languageyes
Author Response
Response to Reviewer 3 Comments
|
||
1. Summary |
|
|
Thank you very much for taking the time to review this manuscript. Please find the detailed responses below and the corresponding revisions/corrections highlighted/in track changes in the re-submitted files. Reviewer comment: 1-)you can improve the title of the study. Author response: Title has been updated. Reviewer comment: 2-)write non-abbreviated version of IDP in the abstract. Author response: This has been done now. Reviewer comment: 3-)you can be careful about punctuation. Author response: Grammatical review completed now. Reviewer comment: 4-)if possible you can give more information about the results in the abstract section. Author response: More information has been added to the abstract now, but with a lot of restriction due to the short word count allowed by this journal for this abstract. Reviewer comment: 5-)you can discuss the findings (line 165) in the context of your study. Author response: Thank you for this. On review, we note that the description given on 165 was unnecessary and we have removed that sentence and only referenced the more general line above. Reviewer comment: 6-)213 reference is missing Author response: Reference added. Reviewer comment: 7-)table 1 seems unclear. Author response: A description has been added on what Table 1 covers now. Reviewer comment: 8-)you can give more information about the participants of the study. Author response: More information has been added in the methodology and discussion section. Reviewer comment: 9-)line 340, 341 reference seems missing. Author response: Noted and added, thanks Reviewer comment: 10-)you can highlight the novelty of your study in the discussion section based on the results of your study. Author response: Two sentences have been added to the start of the discussion elaborating this further. table 1 can be improved in the context of your study. it consists of very little information. Author response: Thank you for this. We decided to keep Table 1 as it is as there are references throughout the introduction of where further information can be found on the programme content is readers are interested. Reviewer comment: 2-)be sure the camp information is correct. if possible support it with references. Author response: Our field team at the time double checked all this camp information. It was not possible to add references at that time, and would be very hard now. But we are very confident of the information. Reviewer comment: 3-)line 340, 341 lacks of reference. Author response: Noted and added, thanks Reviewer comment: 4-)give more information about missing data analysis in your study.
Author response: We mentioned we had very little data and describe what we did to manage it as well as have added a reference for this methodology. Reviewer comment: 5-)table can be improved (page 10) Author response: The table that starts on page 10 was a large undertaking and we hope you can agree that it reflects well the data and references made to it in the discussion of the results. Thanks! Reviewer comment: 6-)you can give more information about how much they earn monthly and mention as a confounding factor in the study. Author response: We did not have information of how much they earn, but we have now added to the discussion on limitations this point as a potential confounding factor
|
Round 2
Reviewer 1 Report
Comments and Suggestions for Authors
While some improvements have been made, the manuscript still contains significant language errors, broken figure references, and problematic causal claims despite acknowledging the lack of a control group. These issues substantially affect the clarity and scientific validity of the article. If the necessary corrections — including thorough language editing, correction of all formatting errors, and careful revision of causal statements — are not completed, I recommend that the manuscript be rejected.
Comments on the Quality of English LanguageThe quality of language in the manuscript is currently not sufficient for publication. Despite some corrections, numerous spelling mistakes, grammatical errors, and awkward phrasings remain throughout the text.
Author Response
We thank the reviewer for taking the time to review our paper once again and provide valuable feedback.
1-We have undergone external proof reading and editing now, and expect no language or grammatical errors to be found.
2-The authors have thoroughly reviewed the paper and edited all sentences in every section to remove any causal references or claims or statements.
3- We have left all track changes, though it is quite messy, we thought this would indicate best the changes we have made.
4-We could not see any broken figure references in the last version we uploaded. This had been corrected in the previous revision. We are not sure how this was still showing in the reviewers last version of the review but hope this is not the case.
Once again, we thank the reviewer for such thorough comments that have allowed us to improve the paper significantly.
Reviewer 3 Report
Comments and Suggestions for Authors
Accept in present form
Author Response
We thank the reviwer for their feedback and wish them well.